# Advanced Prediction of Hepatic Oncogenic Transformation in HBV Patients via RNA-Seq Data Analysis and Deep Learning Techniques

**DOI:** 10.3390/ijms25189827

**Published:** 2024-09-11

**Authors:** Zhengtai Li, Lei Huang, Changyuan Yu

**Affiliations:** College of Life Science and Technology, Beijing University of Chemical Technology, Beijing 100029, China; lizhengtaii@126.com (Z.L.);

**Keywords:** hepatitis B virus, liver cancer, deep learning, RNA-seq

## Abstract

Liver cancer, recognized as a significant global health issue, is increasingly correlated with Hepatitis B virus (HBV) infection, as evidenced by numerous scientific studies. This study aims to examine the correlation between HBV infection and the development of liver cancer, focusing on using RNA sequencing (RNA-seq) to detect HBV sequences and applying deep learning techniques to estimate the likelihood of oncogenic transformation in individuals with HBV. Our study utilized RNA-seq data and employed Pathseq software and sophisticated deep learning models, including a convolutional neural network (CNN), to analyze the prevalence of HBV sequences in the samples of patients with liver cancer. Our research successfully identified the prevalence of HBV sequences and demonstrated that the CNN model achieved an exceptional Area Under the Curve (AUC) of 0.998 in predicting cancerous transformations. We observed no viral synergism that enhanced the pathogenicity of HBV. A detailed analysis of sequences misclassified by the CNN model revealed that longer sequences were more conducive to accurate recognition. The findings from this study provide critical insights into the management and prognosis of patients infected with HBV, highlighting the potential of advanced analytical techniques in understanding the complex interactions between viral infections and cancer development.

## 1. Introduction

Liver cancer remains a global health challenge, with the number of cases worldwide expected to exceed one million by 2025. Infections with Hepatitis B virus (HBV) and Hepatitis C virus (HCV) are considered major risk factors for Hepatocellular Carcinoma (HCC) [1]. Research shows a strong link between Hepatitis B virus (HBV) infection and liver cancer. However, not all HBV carriers develop liver cancer. This discrepancy is particularly noticeable in areas where HBV is widespread [2,3]. Over half of global liver cancer cases are linked to viral infections, including HBV [4]. Annually, HBV and HCC contribute to more than 1.3 million deaths [5].

Numerous studies have explored the replication mechanisms of Hepatitis B virus (HBV), a double-stranded DNA virus. HBV replicates through a unique reverse transcription process involving an RNA intermediate known as pre-genomic RNA (pgRNA) [6,7]. This process, like that of retroviruses, enables the tracking of HBV activity by detecting RNA intermediates. HBV’s genome is notably compact, and its replication mechanism bears similarities to retroviruses [8]. Upon entering hepatocytes, HBV’s relaxed circular double-stranded DNA (rcDNA) is converted into stable covalently closed circular DNA (cccDNA). This cccDNA is then transcribed by the host RNA polymerase II to produce various RNA intermediates [9]. Moreover, the HBV genome contains no non-coding sequences; all regulatory regions, including enhancers and promoters, are integrated within the protein-coding sequences. This indicates a high efficiency of HBV DNA utilization, involving the entire genome in protein production [10,11]. Additionally, the molecular pathology of HCC tumors varies according to different genotoxic insults and etiologies. Current research is ongoing to translate molecular and immunological categories into biomarkers for guiding treatment [1]. These characteristics underscore the unique replication mechanism of HBV, involving reverse transcription of RNA intermediates and translation of its DNA, key factors in HBV pathogenesis, and challenges in detection and treatment. These challenges could potentially be overcome through deep learning models and extensive data analysis.

In the current research, deep learning models have been applied to the analysis of transcriptomic features shared across various types of cancer, involving extensive training of neural networks with a plethora of RNA-seq samples from diverse tissues and tumor types, to differentiate between normal and tumor tissues [12]. Additionally, deep learning models based on individual transcriptomic data have been developed to optimize drug selection for patients with cancer, demonstrating the potential to improve cancer prognosis [13]. Advances in deep learning in cancer pathology have facilitated the direct extraction of key information from complex molecular biology assays, playing a significant role in identifying new oncological biomarkers [14]. RNA sequencing, as a revolutionary method of transcriptomic analysis, has evolved from bulk RNA sequencing to more precise single-molecule, single-cell, and spatial transcriptomics [15,16]. Messenger RNA tests based on RNA sequencing data have emerged as a new diagnostic tool, complementing traditional DNA genetic testing [17]. Recent studies indicate that deep learning models show significant potential in predicting the relationship between HBV and HCC [18,19,20].

However, existing deep learning models predicting HCC from HBV often focus on completed integration sites or are centered around radiological studies requiring extensive manual identification. Such studies encompass multiple sources, inconsistent processing methods, and diverse etiologies, leading to models not performing as anticipated. Previous research often relied on standard reference sequences for sequence learning, but actual applications face challenges with HBV site mutations exceeding the learning scope, significantly increasing data complexity and potentially impairing model predictive performance. To address this, we plan to directly analyze next-generation sequencing transcriptomic data from diverse sources of HCC patients, identifying HBV sequences as research samples. Our goal is to analyze potential tumor-causing HBV residual sequences, learn their features, and apply this knowledge to future high-throughput sequencing data. Based on these sequencing results, we aim to predict whether patients with hepatitis B are at risk of developing liver cancer.

## 2. Results

### 2.1. Using Pathseq to Identify HBV Sequences

In this study, using all projects, we quantified HBV and assessed other viruses in liver cancer samples and control groups. Figure 1A shows the HBV genome coverage in tumor and normal samples. Coverage is plotted on a log scale, highlighting key regions (Core, Pre-Core, S, and X) of the HBV genome. While the overall coverage trends are similar across both groups, there are key differences in specific regions. Tumor samples (blue line) display higher coverage in most of the S region, which is crucial for HBV replication. In contrast, normal samples (red line) show higher coverage in the Pre-Core and X regions. These differences suggest that different regions of the HBV genome are more active or retained in tumor versus normal tissues, supporting the idea that HBV plays a distinct role in liver carcinogenesis. The residual HBV sequences also showed a high number of mutations, with varying degrees of base mutations across different sites. In Appendix A, we used a 30 bp window to display the mutation patterns in tumor samples, and a similar pattern was observed in the control group.

The heatmap in Figure 1B, which standardized the expression levels of various viruses across all tumor samples, identified HBV as the predominant virus, underscoring its potential role as a primary causative factor in most liver cancer cases in our dataset. This finding provides valuable data for further model research.

We also analyzed the main quantities of different viruses in the liver cancer dataset. Figure 1C displays viral expression levels adjusted for gene length and expression count, shown in logarithmic form for clearer graphical presentation. Although HBV was undetectable in some patients with liver cancer—either due to non-infection, undetectable levels in the transcriptome, or complete absence—Figure 1D illustrates the proportions of various viruses within the samples. Combined with the earlier heatmap, the results show that while most samples contained HBV residuals, the proportion varied across samples, indicating the need for further research into the relationships between different viruses.

To explore the relationships between different viruses and the potential for synergistic effects with HBV, we conducted a detailed correlation analysis of the viruses found in the samples and constructed their phylogenetic trees. This analysis helps to understand evolutionary relationships and possible interactions among the viruses, which could inform the integration of these dynamics into our research model.

Figure 2A shows the correlations among the fifteen major viruses identified. Notably, our primary subject of study, HBV, did not exhibit any significant correlation with other viruses. This also indicates that there were no cases of HBV and HCV co-infection in our samples. Previous studies have suggested that co-infection with both viruses can lead to more complex clinical scenarios [21]. The phylogenetic tree analysis in Figure 2B displays the evolutionary relationships of major viruses. HBV appears on a relatively independent branch with minimal association with other viruses, confirming earlier correlation analysis results. Due to its relative isolation, our model development exclusively analyzes HBV sequences.

### 2.2. Establishing a Deep Learning Model Based on HBV Sequences

We classified the viruses in datasets of HBV-infected but non-cancerous samples and confirmed liver cancer samples. We obtained 1,393,240 positive sample sequences and 2,532,180 negative sample sequences. We initially encoded all sequences using one-hot encoding and then applied the t-SNE method to reduce dimensionality, revealing the characteristics and distribution patterns of these sequences as illustrated in the two images of Figure 3. Directly displaying all sequences would render the images indistinct; therefore, we randomly sampled 500 sequences from each group for plotting. Due to space limitations in this article, we present only two of these images; additional results are available in Appendix A. As can be observed from the images, the two groups of sequences are intermingled, indicating the necessity for a more advanced model to separate them effectively.

Next, we employed deep learning techniques to learn the features of the sequences. The specific structure of the model is depicted in Figure 2, while its performance is detailed in Table 1 and Table 2, and Figure 4.

The CNN model showed the best classification performance in this process. The Transformer model also delivered good results. However, the RNN model demonstrated significantly low recall, indicating weaker performance in classifying positive samples. This issue might stem from an imbalance between positive and negative samples in our dataset, as RNN models often need a more balanced dataset.

### 2.3. Study of Misclassified Sequences

Upon analyzing the sequences misclassified by our top-performing CNN model, we observed that a significant proportion of these sequences were notably shorter in length, with many falling below 100 bp. This shorter length resulted in insufficient information for the model to effectively learn the distinguishing features, leading to misclassification. Specifically, nearly 40% of the false positive sequences and over 80% of the false negative sequences were under 100 bp, as illustrated in Figure 5A,B. To address this issue, we applied zero-padding during data processing to standardize the sequence length, which allowed us to include more sequences in the model. Despite this, many misclassified sequences were approximately half the specified 145 bp length. Future improvements to the model could involve more sophisticated data preprocessing techniques to better manage variability in sequence coverage and length, potentially reducing misclassification rates.

## 3. Discussion

In this study, we identified substantial HBV residuals in patients with liver cancer using RNA-seq data. Our analysis shows that HBV is the most prevalent virus in tumor samples, strongly indicating its central role in liver cancer development compared to other viral infections. This reinforces the direct association between HBV and liver cancer. Our correlation analysis further demonstrated that viruses in individuals with HBV- were relatively independent, lacking strong synergistic effects with other viruses, which led us to focus solely on HBV sequences for predicting cancerous changes.

The CNN model demonstrated superior performance with an AUC of 0.998, highlighting its potential for real-world applications, particularly in the early detection of liver cancer in individuals with HBV. Compared to traditional methods that rely on manual identification of viral integration sites or radiological data, our CNN-based approach automates the detection of HBV sequences from RNA-seq data. This automation not only improves speed and accuracy but also reduces human error, making the model particularly useful in clinical workflows, especially in regions with high HBV prevalence where early screening is critical.

However, variability in sequence length and coverage posed challenges for the model. Many misclassified sequences were shorter than 100 bp, making it difficult for the model to capture distinguishing features. While zero-padding helped standardize sequence lengths, further improvements in handling this variability will enhance the model’s performance. Such variability is inevitable when working with raw RNA-seq data from non-human sequences.

Future work will focus on experimental validation of the model’s predictions and the development of an online prediction platform. Additionally, the discovery of multiple viruses, such as Epstein–Barr virus (EBV), in patients with liver cancer suggests a need for further exploration of their potential role in liver cancer progression. Integrating gene expression profiles into future models may offer deeper insights into the interactions between viral infections and cancer.

## 4. Materials and Methods

### 4.1. Extracting HBV Sequences

In this research, we utilized RNA-seq data from three studies to generate input files for training the model. These studies include one liver cancer sample project, PRJNA867011, and two control group projects studying HBV infection: liver fibrosis (PRJNA946157) [22] and HBV infection (PRJNA933084) [23]. The specific information for all items is listed in Table 3.

The HBV sequences extracted from our datasets exhibited variability in coverage and length. Shorter sequences and those with mutations across different sites posed challenges for the model. To address this, we applied zero-padding to standardize the lengths of sequences, ensuring consistency across the dataset. However, regions with lower coverage or high mutation rates remained a factor that impacted the model’s ability to generalize well. This variability in coverage and sequence length, especially for shorter sequences, may influence the classification performance, with a higher rate of misclassification observed in these cases. Importantly, the variability in sequence length is an inherent and unavoidable issue in future model applications, as we are processing non-human sequence fragments from raw sequencing data, which naturally exhibit this heterogeneity.

We used project PRJNA867011 to demonstrate the quantification and distribution of the Hepatitis B virus in RNA-seq data, while the other samples are used in the process of building the final deep learning model. The Pathseq tool of GATK [24] (Version 4.2.0) was employed to identify HBV sequences within the samples, with host sequences aligned and removed using the GRch38 reference. After HBV sequence identification, the remaining sequences were classified. We consolidated the viral expression levels of all samples together, computed the Transcripts Per Million (TPMs) for the final expression matrix as the standardized final table, and performed visualization. The processing of Pathseq result files and the correlation analysis with various viruses were conducted using R (Version 4.2.2). The construction of the phylogenetic trees was performed using MEGA [25] (Version 11.0.13), and the trees were visualized using iTOL (https://itol.embl.de/ accessed on 11 December 2023) [26]. Based on the patients’ final status, sequences were categorized into two groups: normal and tumor. Finally, using Samtools (Version 1.16.1) [27] in Linux, we organized all the sequences into two fasta files suitable for input into the model. The entire analytical process is depicted in Figure 6.

### 4.2. Deep Learning Environment and Data Preprocessing

During the development of the deep learning models, Pytorch (Version 2.0.0) [28] was utilized for model construction. In the CNN and RNN models, sequences were transformed into one-hot encodings, whereas in the Transformer model, sequences were directly mapped numerically and used as input. Given the substantial size of the dataset, it was divided into training, validation, and final test sets in a 4:3:3 ratio, respectively.

### 4.3. Deep Learning Models

In this study, we selected convolutional neural network (CNN), recurrent neural network (RNN), and Transformer models due to their well-established strengths in biological sequence analysis. CNNs are widely recognized for their ability to capture local dependencies and patterns in sequence data, making them particularly suited for analyzing the compact HBV genome and identifying short-range relationships or structural features within viral sequences. RNNs, on the other hand, are commonly applied to biological data for their capacity to model sequential information and capture dependencies across the entire sequence, which helps reveal potential associations that span larger regions of the HBV genome. The Transformer model, with its self-attention mechanism, efficiently handles long sequences, capturing both short- and long-range dependencies. Its capacity to process large datasets and learn complex interactions within the HBV genome makes it particularly valuable for this analysis. The selection of these three models was based on their proven effectiveness in biological sequence analysis, enabling the accurate prediction of oncogenic transformations from HBV sequences.

#### 4.3.1. Convolutional Neural Network

In this study, sequence data analysis was conducted using a one-dimensional convolutional neural network (1D CNN). The model’s architecture includes a one-dimensional convolutional layer (Conv1d) and a fully connected layer (Linear). The Conv1d layer features 4 input channels and 32 output channels and uses a kernel size of three and a stride of one. The convolutional process of this layer is described by the following mathematical formula:
(1)$$c_{i}=\sum_{j=1}^{3}f_{j}\cdot x_{i+j-1}$$where x represents the input sequence, f is the convolution kernel, and $c_{i}$ is the convolution output at position i. Subsequently, the model employs ReLU as a nonlinear activation function to enhance its capability for nonlinear representation. The fully connected layer maps the feature output from the convolutional layer (flattened to a dimension of 2272) to two output classifications, making it suitable for binary classification tasks. The linear transformation in this layer can be expressed as follows:
(2)y=Wx+b
where W and b are the weight matrix and bias term, respectively.

During the training phase, we used the cross-entropy loss function to measure the model’s predictive accuracy. Simultaneously, we adjusted parameters using the Adam optimization. The architecture of the CNN model is detailed in Figure 7A.

#### 4.3.2. Transformer Network

Subsequently, we implemented the Transformer model to process sequence data, a framework designed to capture long-range dependencies in sequences. The core of this model includes an embedding layer that projects each sequence element into a high-dimensional space, effectively capturing complex features. The output from the embedding layer is then enhanced by a positional encoder, which uses a combination of sine and cosine functions to provide unique encodings for each position. The specific mathematical expressions used are outlined as follows:
(3)$$PE\left(pos,2i\right)=\sin\left(\frac{pos}{10000^{\frac{2i}{d_{\mathrm{model}}}}}\right)$$
(4)$$PE\left(pos,2i+1\right)=\cos\left(\frac{pos}{10000^{\frac{2i}{d_{\mathrm{model}}}}}\right)$$

The fundamental architecture of the Transformer model is composed of an encoder and a decoder, each leveraging a self-attention mechanism. Within the self-attention layer, the input transforms three distinct components: queries (Q), keys (K), and values (V). This is followed by the computation of attention scores, which are mathematically expressed as follows:
(5)$$\mathrm{Attention}\left(Q,K,V\right)=\mathrm{softmax}\left(\frac{QK^{T}}{\sqrt{d_{k}}}\right)V$$

The self-attention mechanism in the model allows it to process all sequence elements at once, capturing complex relationships. The encoder layer includes a feed-forward network, layer normalization, and Dropout to improve stability and generalization. The decoder layer, like the encoder, adds another attention mechanism to merge its output with the encoder’s. Finally, a fully connected layer transforms the Transformer’s output into the final predictive output.

During the training phase, the cross-entropy loss function was utilized as the primary means to optimize the model, alongside the implementation of the AdamW optimizer for the adjustment of model parameters. The specific structure of the Transformer model is illustrated in Figure 7C.

#### 4.3.3. Recurrent Neural Network

Ultimately, a recurrent neural network (RNN) model was employed for the processing and analysis of sequence data. The recurrent layer is designed to accommodate four input features and contains 64 hidden units, thereby effectively addressing the time-dependent relationships present in sequence data. The fundamental mathematical principle underpinning the RNN model is articulated as follows:

For the hidden state $h_{t}$ at time step t, it is computed from the previous time step’s hidden state $h_{t-1}$ and the current time step’s input $x_{t}$ through an activation function f:
(6)$$h_{t}=f\left(W_{hh}h_{t-1}+W_{xh}x_{t}+b_{h}\right)$$
where $W_{hh}$ is the weight matrix of the hidden layer itself, $W_{xh}$ is the weight matrix from the input to the hidden layer, and $b_{h}$ is the bias term of the hidden layer. The output of the RNN layer $h_{t}$ is then passed to a fully connected layer for the final classification task. The fully connected layer maps the output of the RNN layer to two output features, suitable for binary classification tasks. The linear transformation in this layer can be represented as follows:
(7)$$y_{t}=W_{hy}h_{t}+b_{y}$$
where $y_{t}$ is the output at time step t, $W_{hy}$ is the weight matrix from the hidden layer to the output layer, and $b_{y}$ is the bias term of the output layer. Additionally, to adjust the model parameters, we employed the Adam optimizer, which combines momentum updates and adaptive learning rate techniques. The specific structure of the RNN model is illustrated in Figure 7B.

## 5. Conclusions

In this study, we developed and applied a method to analyze RNA-seq data for detecting residual HBV sequences in patients with liver cancer. Our approach effectively identified HBV as the most prevalent virus in tumor samples, and through correlation analysis, we found that HBV was mutually exclusive from other viral infections in these patients, highlighting its central role in liver cancer development.

The analysis tested three models well suited for sequence analysis: CNN, RNN, and Transformer. Among these, the CNN model achieved the highest accuracy with an AUC of 0.998. However, we found that sequence length significantly impacted classification performance. Specifically, sequences shorter than 100 bp were more prone to misclassification, with nearly 40% of false positives and over 80% of false negatives occurring in this range. Despite zero-padding to standardize sequence lengths, variability in coverage remained a challenge for the model.

This research demonstrates the potential of deep learning techniques in understanding HBV-related liver cancer and highlights the importance of precise sequence analysis for improving disease prediction and diagnosis.

## Figures and Tables

**Figure 1 ijms-25-09827-f001:**
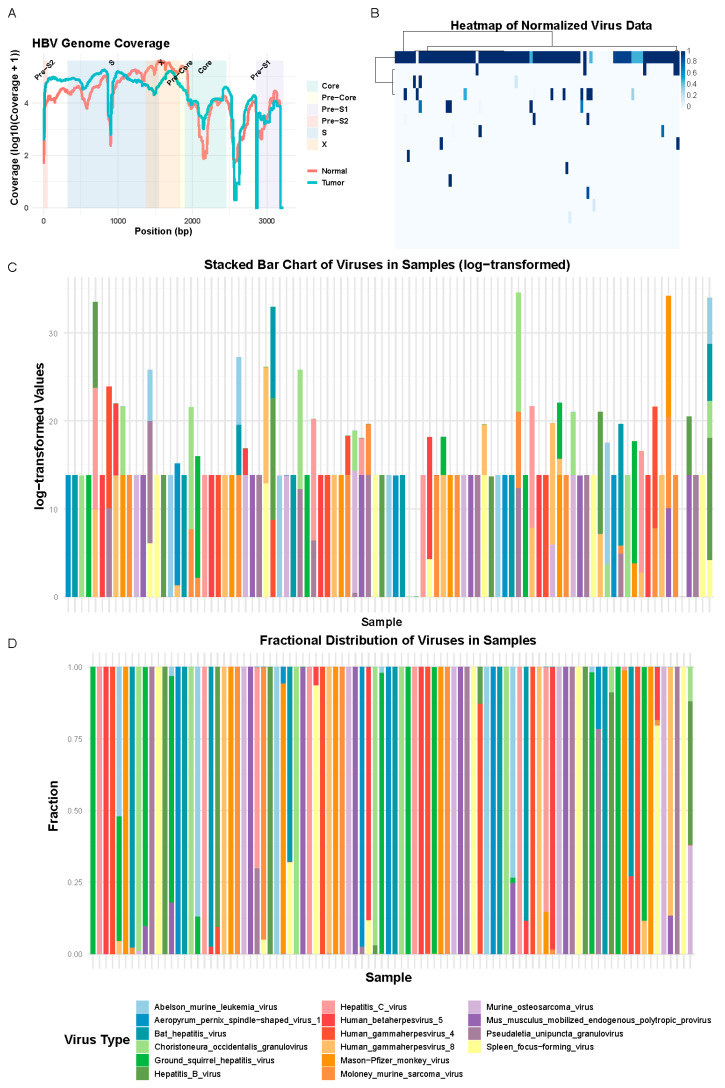
Figure (**A**) displays the coverage of detected HBV sequences in liver cancer samples and normal samples on the standard HBV sequence. Figure (**B**) is a heatmap of the contents of all viruses. Figure (**C**) shows the quantity of each type of virus in each sample, while Figure (**D**) presents the proportion of each type of virus in each sample as a percentage.

**Figure 2 ijms-25-09827-f002:**
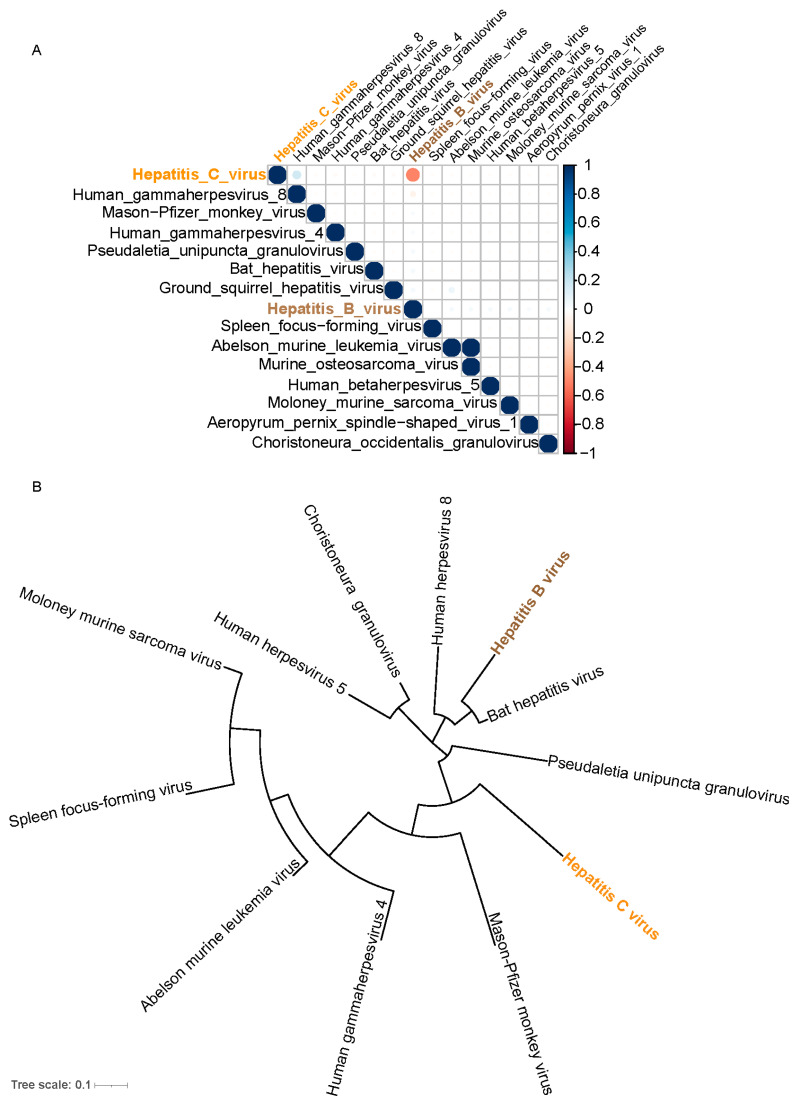
Figure (**A**) displays the correlations among the various viruses we identified. Figure (**B**) illustrates the phylogenetic relationships between these viruses.

**Figure 3 ijms-25-09827-f003:**
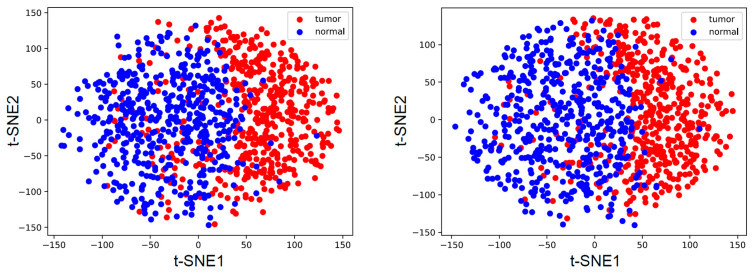
Results after dimensionality reduction using t-SNE method on two randomly selected groups of one thousand sequences each.

**Figure 4 ijms-25-09827-f004:**
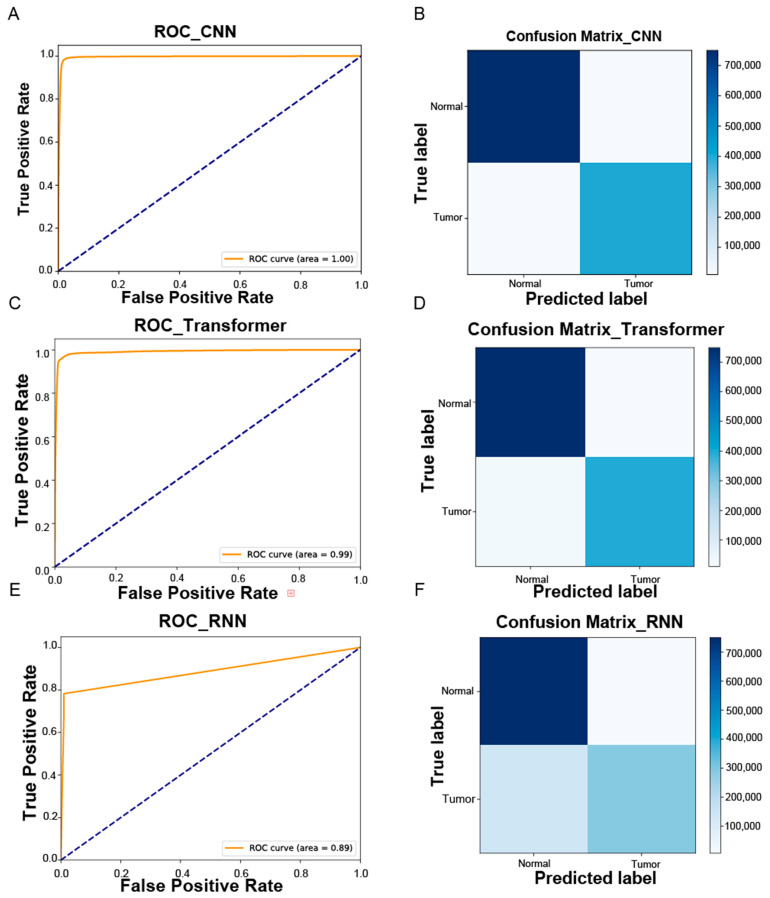
Figures (**A**,**B**) illustrate the ROC curve and the confusion matrix using the CNN model. Figures (**C**,**D**) display the ROC curve and the confusion matrix for the Transformer model, respectively. Figures (**E**,**F**) show the ROC curve and the confusion matrix for the RNN model, respectively.

**Figure 5 ijms-25-09827-f005:**
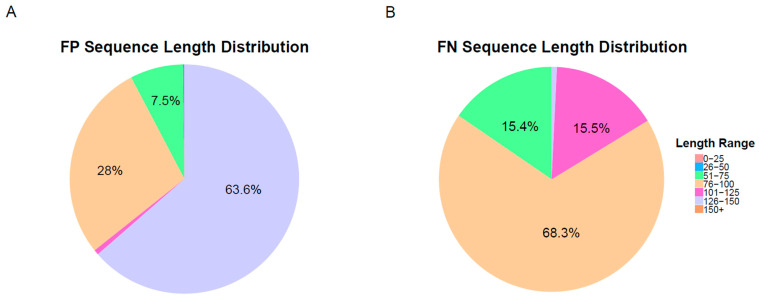
Figure (**A**) displays the length distribution of false positive sequences, while Figure (**B**) shows the length distribution of false negative sequences.

**Figure 6 ijms-25-09827-f006:**
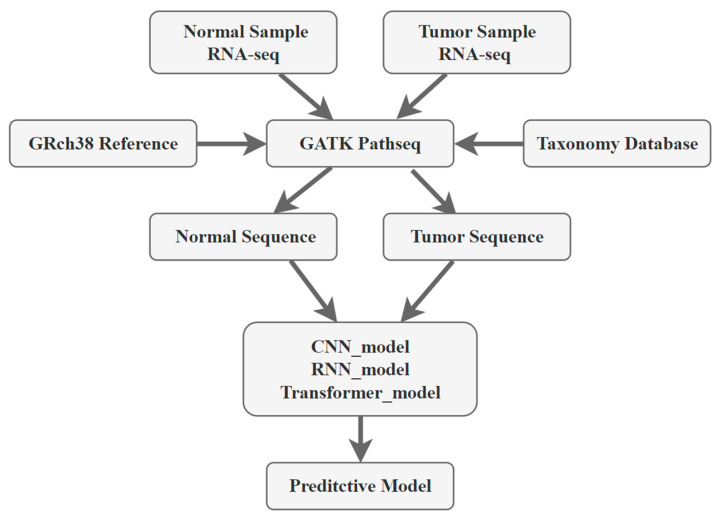
The overall process of this study.

**Figure 7 ijms-25-09827-f007:**
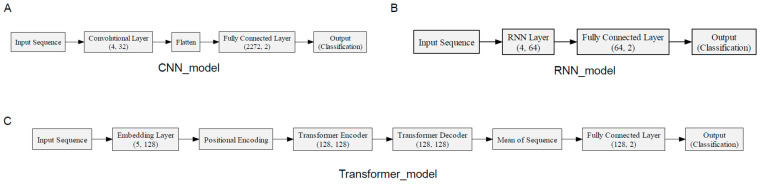
Figure (**A**) displays the structure and parameter settings of the CNN model, Figure (**B**) shows the structure and parameters of the RNN model, and Figure (**C**) presents the structure and parameters of the Transformer model.

**Table 1 ijms-25-09827-t001:** Performance of the model on the validation dataset.

Models	Accuracy	Precision	Recall	F1 Score
HBV_CNN	0.981	0.975	0.978	0.972
HBV_Transformer	0.973	0.973	0.951	0.960
HBV_RNN	0.883	0.973	0.688	0.798

**Table 2 ijms-25-09827-t002:** Performance of the model on the test dataset.

Models	Accuracy	Precision	Recall	F1 Score	AUC
HBV_CNN	0.981	0.975	0.971	0.972	0.998
HBV_Transformer	0.973	0.972	0.950	0.959	0.991
HBV_RNN	0.883	0.974	0.689	0.798	0.887

**Table 3 ijms-25-09827-t003:** Sample information.

Project Number	Year	Sample Type	Number of Samples
PRJNA867011	2022	Liver tissue	98 Liver Cancer
PRJNA946157	2023	Liver tissue	5 Liver Fibrosis
PRJNA933084	2023	Serum	21 Humans with HBV

## Data Availability

The program used in this article for identifying HBV sequences and the subsequent script for deep learning have all been uploaded to the following URL: https://github.com/LiZhengTai2022/HBV_DL (accessed on 15 February 2024).

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
