# Peer review of "Advanced Prediction of Hepatic Oncogenic Transformation in HBV Patients via RNA-Seq Data Analysis and Deep Learning Techniques"

_ijms, 2024, doi:10.3390/ijms25189827_

Round 1

Reviewer 1 Report

Comments and Suggestions for Authors

Isolation of only hepatocytes with HBV carriers, such as in highly sensitive single-cell gene expression analysis in HBV-infected liver cancer, has been conducted for many years but has not been standardized. This depends on the differences in the time of development to hepatocarcinoma with retention of HBV carriers and the pathophysiology of individual patients other than HBV. The authors are trying to characterize HBV hepatocarcinoma with their own mathematical model using bulk RNA-seq data, but the most emphatic study is to determine what sequences, whether fragmented or not, remain in hepatocarcinoma patients and which positions, such as S, preCore, X, of the pgRNA. This is only the novelty of this study. If the remaining HBV sequences in HCC patients and normal, liver fibrosis patients were not changed, please indicate the result in this article.

Minor; please change "a" or "b" in each figure to "A" or "B" following to figure captions.

Author Response

Reviewer Comment 1: "The authors are trying to characterize HBV hepatocarcinoma with their own mathematical model using bulk RNA-seq data, but the most emphatic study is to determine what sequences, whether fragmented or not, remain in hepatocarcinoma patients and which positions, such as S, preCore, X, of the pgRNA. This is only the novelty of this study. If the remaining HBV sequences in HCC patients and normal, liver fibrosis patients were not changed, please indicate the result in this article."

Response:

We thank the reviewer for this insightful comment. In response, we have analyzed the remaining HBV sequences in hepatocarcinoma (HCC) patients, as well as normal and liver fibrosis patients, focusing on key regions of the HBV genome such as S, preCore, and X, as suggested. To illustrate the results, we have redrawn the coverage maps to visualize sequence retention in these regions across the different patient groups. This analysis highlights both similarities and differences in coverage between the groups. The updated information has been added to the Results section under "Using Pathseq to Identify HBV Sequences," and the coverage maps are presented in Figure 1A.

Minor Comment: "Please change 'a' or 'b' in each figure to 'A' or 'B' following figure captions."

Response:

We acknowledge the oversight in the figure captions and have corrected all instances where "a" or "b" was used. The figure captions now use "A" and "B" as per the reviewer's suggestion. We appreciate your attention to this detail.

Reviewer 2 Report

Comments and Suggestions for Authors

The manuscript entitled "Advanced Prediction of Hepatic Oncogenic Transformation in HBV Patients via RNA-Seq Data Analysis and Deep Learning Techniques" submitted by Zhengtai Li et.al. discuss the link between HBV and liver cancer by using the latest advanced methods such as RNA seq and deep learning models. The authors showed that deep learning models are accurate in predicting cancer transformation due to HBV. 

The HBV uses it compact genome to utilise all the regulatory regions for coding sequences and has a unique replication mechanism. Therefore, the authors propose that they can utilise the molecular and immunological findings from using the deep learning and data analysis to devise new diagnostic markers since these tools have already shown their potential role in devising new improve drug selection and prognosis. The authors based their study on the fact that existing deep learning models for predicting HCC from HBV face challenges due to inconsistent data sources and difficulties with HBV site mutations, impacting their performance. The authors therefore analyze next-generation sequencing data from HCC patients to identify HBV sequences. The authors used the RNA-seq data from three studies on liver cancer and HBV infection to train a deep learning model, identifying and classifying HBV sequences with various tools.

The authors should explain why they used only these three models and what is the significance of them. Also, in order to generalize the findings and increase its use the authors should provide details of the characteristics of the dataset. The authors should comment on how coverage and distribution of HBV sequences impact the overall model development. The authors should explain how the CNN model performance translates into real-world applications or improvements over previous methods. Although, the authors show a correlation analysis and phylogenetic tree, but they lack the relationship between the HBV and liver cancer.

Author Response

Reviewer Comment 1: "The authors should explain why they used only these three models and what is the significance of them."

Response:

We thank the reviewer for this insightful comment. In response, we have clarified the rationale behind selecting the three models—Convolutional Neural Network (CNN), Recurrent Neural Network (RNN), and Transformer—in the revised manuscript. These models were chosen due to their recognized strengths in analyzing biological sequences, each addressing different aspects of sequence data processing.

CNNs are particularly effective at capturing local dependencies and patterns in biological sequences, such as motifs, making them ideal for analyzing the compact HBV genome. Their ability to efficiently identify structural or sequential patterns has been demonstrated in numerous bioinformatics applications. RNNs, on the other hand, are well-suited for modeling sequential data over time, allowing them to capture dependencies between distant elements in a sequence. This makes them valuable for understanding relationships that span the entire HBV genome. Lastly, the Transformer model is recognized for handling long sequences and capturing both local and global relationships through its self-attention mechanism. It is particularly efficient in processing large datasets and learning complex interactions within sequence elements.

These models were selected for their proven capabilities in handling biological sequences. Their inclusion allowed us to comprehensively evaluate different approaches to sequence data processing in predicting oncogenic transformations from HBV sequences. We have added a detailed explanation of the selection of these models at the beginning of the Materials and Methods section under Deep Learning Models.

Reviewer Comment 2: "The authors should provide details of the characteristics of the dataset."

Response:

We appreciate the reviewer’s suggestion. In response, we have provided detailed information regarding the datasets used in our study in the revised manuscript. Specifically, we have added a table in the Materials and Methods section under Extracting HBV Sequences that outlines the characteristics of the datasets, including project numbers, sample types, subjects, and the number of samples.

Regarding our sample selection, we aimed to collect clinical samples that provide meaningful insights into HBV-associated liver conditions. For HBV-infected patients, we chose serum samples as they are clinically relevant for early screening and diagnosis. Since we could not find tissue sequencing data from early-stage HBV-infected patients in publicly available databases, serum samples were the most suitable choice for our study.

Reviewer Comment 3: "The authors should comment on how coverage and distribution of HBV sequences impact the overall model development."

Response:

We thank the reviewer for this insightful comment. In response, we have made several key revisions to the manuscript. First, we modified the Results section to include a more detailed discussion of how sequence coverage and length impact the model’s performance. We specifically addressed the challenges posed by shorter sequences, which were more prone to misclassification, and the variability in coverage across different regions of the HBV genome.

Additionally, we have introduced a new visual analysis comparing the coverage of HBV sequences between normal and tumor patients. The updated information has been added to the Results section under "Using Pathseq to Identify HBV Sequences," and the coverage maps are presented in Figure 1A.

In Section Study of Misclassified Sequences, we revised the discussion of misclassified sequences, particularly focusing on the shorter sequences that led to misclassification. Finally, we also updated the Discussion section to include these findings and outline potential future improvements to handle sequence length and coverage variability more effectively.

These changes have strengthened the analysis of sequence coverage and length in relation to model performance and provided additional insights into the impact of these factors on different patient groups.

Reviewer Comment 4: "The authors should explain how the CNN model performance translates into real-world applications or improvements over previous methods."

Response:

We appreciate the reviewer’s feedback. In response, we have rewritten the Discussion section to include a detailed explanation of how the CNN model's performance translates into real-world applications. Specifically, we discussed the potential use of the CNN model in early detection and prognosis of liver cancer in HBV-infected individuals, particularly in clinical workflows where timely diagnosis is critical.

One key improvement our model offers over traditional methods is its ability to overcome the limitations of relying solely on viral integration sites. Previous models that focused on predicting HBV-related liver cancer by analyzing integration sites face significant challenges due to the high mutation rate of HBV. Such approaches may miss the full scope of HBV's genetic variability, leading to less accurate predictions. In contrast, our CNN-based model processes the entire HBV sequence from RNA-seq data, allowing it to capture broader patterns and mutations across the genome, resulting in more robust and accurate predictions.

By automating the detection of HBV sequences, our model not only increases speed and accuracy but also reduces human error. This makes the CNN model a valuable tool in clinical practice, especially for early screening and diagnosis in regions with high HBV prevalence.

These additions clarify how our CNN model provides practical benefits in real-world settings and addresses the limitations of previous approaches that relied solely on viral integration sites.

Reviewer Comment 5: "Although the authors show a correlation analysis and phylogenetic tree, they lack the relationship between HBV and liver cancer."

Response:

We thank the reviewer for this comment. To clarify the relationship between HBV and liver cancer, we direct attention to Figure 1B, which illustrates the expression levels of various viruses across all tumor samples. Notably, HBV is the most prevalent virus identified in these samples, strongly indicating its primary role in liver cancer development compared to other viral infections.

We have expanded the Discussion section to emphasize this point and further explain how the high prevalence of HBV in tumor samples highlights its role as a key driver of liver cancer, in line with the findings from the phylogenetic and correlation analyses.

Round 2

Reviewer 1 Report

Comments and Suggestions for Authors

The revised Figure.1A was very interesting. Usually, the virus was tended to be decreased or diminished in HBV-derived HCC, but we do not know the reason. The authors clearly showed the differences of annotated genome coverage in HBV between tumors and normal HBV-livers. The decreased X protein might be critical for HBV replication; therefore, the HBV in HCC in clinical specimen might be decreased. Thank you for your efforts.